# Regiodivergent biosynthesis of bridged bicyclononanes

Lukas Ernst [1] ✉, Hui Lyu [2], Pi Liu [3], Christian Paetz [2], Hesham M. B. Sayed[1,4], Tomke Meents [1], Hongwu Ma [3], Ludger Beerhues [1,5], Islam El-Awaad [1,4,5] ✉ & Benye Liu [1,5] ✉

Medicinal compounds from plants include bicyclo[3.3.1]nonane derivatives, the majority of which are polycyclic polyprenylated acylphloroglucinols (PPAPs). Prototype molecules are hyperforin, the antidepressant constituent of St. John's wort, and garcinol, a potential anticancer compound. Their complex structures have inspired innovative chemical syntheses, however, their biosynthesis in plants is still enigmatic. PPAPs are divided into two subclasses, named type A and B. Here we identify both types in *Hypericum sampsonii* plants and isolate two enzymes that regiodivergently convert a common precursor to pivotal type A and B products. Molecular modelling and substrate docking studies reveal inverted substrate binding modes in the two active site cavities. We identify amino acids that stabilize these alternative binding scenarios and use reciprocal mutagenesis to interconvert the enzymatic activities. Our studies elucidate the unique biochemistry that yields type A and B bicyclo[3.3.1]nonane cores in plants, thereby providing key building blocks for biotechnological efforts to sustainably produce these complex compounds for preclinical development.

The medicinal value of plants of the Hypericaceae and Clusiaceae families is linked to a variety of bioactive meroterpenes, which arise from the conjunction of terpene and polyketide biosyntheses. The meroterpenes include polycyclic polyprenylated acylphloroglucinols (PPAPs), a diverse molecular family, which is chemically characterised by the complex bicyclo[3.3.1]nonane scaffold. Since the first report of a PPAP (hyperforin) in 1971[1,2], this class has rapidly expanded as contemporary research continues to unravel new members with distinct structural elements[3]. The acyl group of PPAPs is either aliphatic or aromatic. PPAPs bearing the benzoyl group, i.e., phlorbenzophenone derivatives, account for the largest proportion of elucidated structures[4,5]. They are abundant in the traditional Chinese medicinal plant *Hypericum sampsonii*[6]. Clusianone and nemorosone, alongside their C-7 epimers, are among the most intensively studied PPAPs with a phlorbenzophenone core (Fig. 1a). Their established in vitro medicinal properties feature anti-viral[7,8], anti-microbial[9,10], anti-inflammatory[11], and cytotoxic activities[11,12]. Similar bioactivities are exerted by their caged derivatives, hyperibone K and plukenetione A, respectively[8,13,14]. Aiming at harnessing their potential as valuable pharmacophores, pioneering studies demonstrated the total syntheses of these compounds in the previous decade[15–18]. Yet, innovative efforts to improve the access to the coveted bicyclo[3.3.1]nonane scaffold continue to impact the recent literature of organic chemistry[19–21].

Despite the promising pharmacological outlook, biosynthesis of the complex PPAPs is largely obscure. Important milestones at the gene level were the assembly of the basic phloroglucinol core[22] and the detection of the first plant-derived aromatic prenyltransferase (PT)[23]. However, the nature of the enzymatic machinery that catalyses the

[1]Technische Universität Braunschweig, Institute of Pharmaceutical Biology, Braunschweig, Germany. [2]Max Planck Institute for Chemical Ecology, NMR/Biosynthesis Group, Jena, Germany. [3]Chinese Academy of Sciences, Tianjin Institute of Industrial Biotechnology, Biodesign Center, Key Laboratory of Engineering Biology for Low-carbon Manufacturing, Tianjin, China. [4]Assiut University, Faculty of Pharmacy, Department of Pharmacognosy, Assiut, Egypt. [5]Technische Universität Braunschweig, Center of Pharmaceutical Engineering, Braunschweig, Germany. ✉e-mail: lukas.ernst@tu-braunschweig.de; islam.elawaad@tu-braunschweig.de; b.liu@tu-braunschweig.de

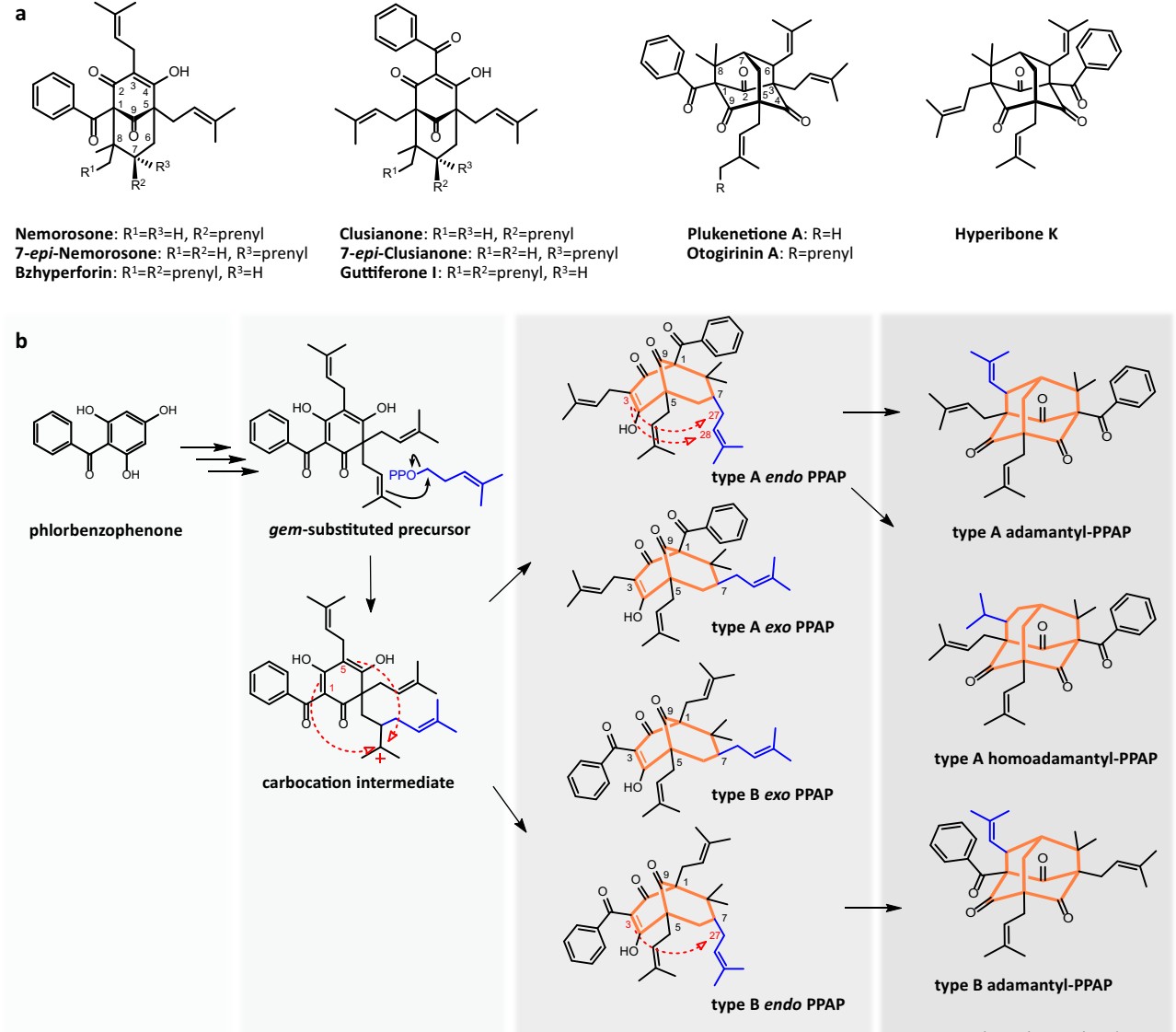

**Fig. 1 | Proposed biosynthesis of phlorbenzophenone-derived PPAPs.**
**a** Examples of PPAPs originating from phlorbenzophenone. **b** Hypothetic scheme depicting the biogenic conversion of phlorbenzophenone to bridged polycyclic derivatives[4]. Red dashed arrows indicate ring closure reactions. The newly added prenyl group and the polycyclic ring systems are highlighted in blue and orange, respectively. All homoadamantane-type PPAPs so far isolated have type A structures.

formation of the bicyclo[3.3.1]nonane skeleton remained unanswered. The following hypothesis for the biosynthetic assembly of the bridged polycyclic compounds was proposed[4,24,25] (Fig. 1b). After polyprenylation of the phlorbenzophenone scaffold catalysed by aromatic PTs, one of the geminal prenyl side chains is activated by head-to-middle prenylation, resulting in the formation of a reactive carbocation intermediate. Subsequent ring closure between this carbocation and either the C-1 or the C-5 position via electrophilic addition yields the bridged bicyclic type A and B scaffolds, respectively. Whether these reactions are catalysed by PTs with additional cyclase functionality ('prenylative cyclization') or by PTs in concert with independent cyclases remained ambiguous. The position of the introduced C-7 prenyl group is either *endo* or *exo* with respect to the C-9 carbonyl. However, only the *endo* configuration is thought to be involved in the formation of metabolites with rare adamantane and homoadamantane scaffolds, which appear to be achieved by intramolecular cyclization between C-3 and C-27 or C-28, respectively[24]. An array of such caged tricyclic PPAPs was identified in *H. sampsonii*[6,24].

Here we detect and characterise two bifunctional PTs from *H. sampsonii*, which catalyse regiodivergent prenylative cyclizations of the same *gem*-diprenylated phorbenzophenone substrate to yield the type A and B products 7-*epi*-nemorosone and 7-*epi*-clusianone, respectively. Substitution of a prenyl with a geranyl group results in the formation of nemosampsone, the undescribed but anticipated geranylated type A precursor of many bi- and tricyclic PPAPs in *H. sampsonii*. We use a combination of computational and mutational studies to investigate the underlying reaction mechanisms. Inverted substrate binding modes control the regiospecific reactions in the two active site cavities. Identification of a set of amino acid residues that stabilise the alternative binding modes allows for reciprocal mutagenesis and results in the conversion of the type A-forming enzyme into a type B cyclase by ninefold mutation. Furthermore, we use the sequence information obtained to identify a homologous enzyme in *Hypericum perforatum*, which forms hyperforin and secohyperforin from substrates bearing an aliphatic rather than aromatic acyl group.

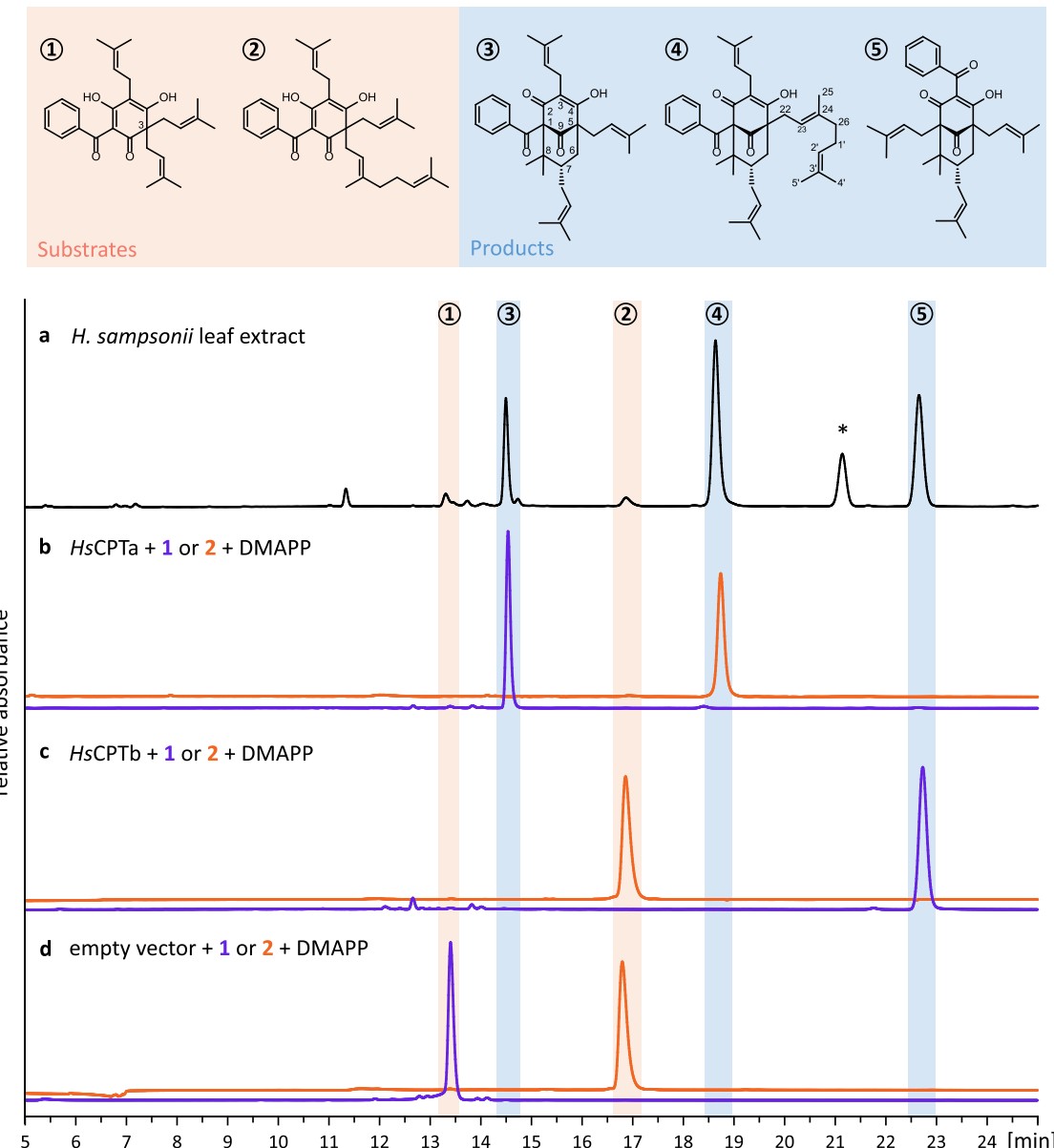

**Fig. 2 | Elucidation of plant constituents and enzymatic products. a** HPLC-DAD analysis of a dichloromethane extract from *H. sampsonii* leaves. **b** Enzyme assays containing *Hs*CPTa, DMAPP, and either grandone **1** (blue chromatogram) or kolanone **2** (orange chromatogram) as acceptor substrates yielded the products 7-*epi*-nemorosone **3** and nemosampsone **4**, respectively. **c** Enzyme assays containing *Hs*CPTb, DMAPP, and grandone **1** (blue chromatogram) yielded 7-*epi*-clusianone **5**. Kolanone **2** (orange) was not transformed. **d** No conversions were observed in control assays containing microsomes from empty vector-transformed yeast cells. The enzyme assays are shown at their respective optimal absorbance wavelengths between 272–360 nm, whereas the leaf extract chromatogram was recorded at 292 nm. *, hypericin (naphthodianthrone).

## Results and discussion

### *H. sampsonii* plants contain type A and B PPAP prototypes

Reversed-phase high-performance liquid chromatography (RP-HPLC) of dichloromethane extracts from fresh leaf material of in vitro propagated and pot-cultivated *H. sampsonii* plants revealed a tractable signal landscape consisting of four major constituents and a few minor metabolites (Fig. 2a). Isolation and NMR spectroscopic elucidation disclosed the structures of the well-studied type A and B PPAP congeners 7-*epi*-nemorosone **3** and 7-*epi*-clusianone **5** (Supplementary Figs. 1–2). Interestingly, the analysis of another major constituent led to the discovery of a yet undescribed PPAP. The 1D and 2D NMR spectra of this compound closely resembled those of **3**, except for the signals of an additional prenyl group at C-26. Because of this similarity, the compound is hereafter referred to as nemosampsone **4**. The

complete elucidation data of **4** is available in the Supplementary Information (Supplementary Table 1 and Supplementary Figs. 3–10). The fourth major constituent exhibited absorption in the visible spectrum, which was characteristic of hypericin. This naphthodianthrone is typical of *Hypericum* species but was not further analysed here. Crucially, examination of the minor extract components identified the putative direct precursors of the aforementioned PPAPs, namely grandone **1** and kolanone **2** (Supplementary Figs. 11–12). These intermediates were utilised as acceptor substrates in subsequent PT assays.

Interestingly, all PPAPs elucidated here, including the newly described compound **4**, were type A and B prototypes without further chemical functionalisation. In contrast, many PPAPs reported for *H. sampsonii* comprise secondary ring closures, which were hypothesised

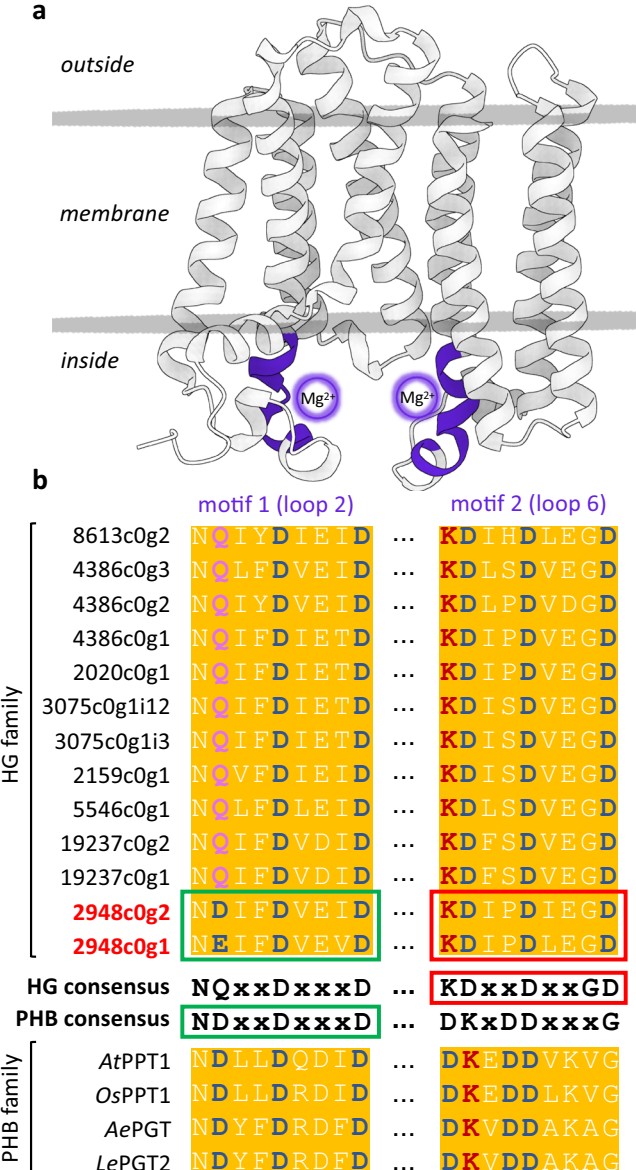

**Fig. 3 | Alignment of conserved aspartate-rich regions of PTs. a** Representative PT topology with highlighted metal ion binding motifs (aspartate-rich regions). **b** All transcriptomically selected candidates had the expected canonical sequences of homogentisate (HG) PTs, except for two (red), which contained an unusual combination of a *p*-hydroxybenzoate (PHB)-like sequence as the first motif and a HG-like sequence as the second. The PHB enzymes listed are examples of previously studied family members.

Previous rigorous NMR studies pointed out that close examination of $^1$H and $^{13}$C shifts provides valuable information about the relative configuration of the C-7 prenyl group of PPAPs[27]. According to the $^{13}$C shifts of C-7 and the geminal methyl groups attached to C-8, along with the $^1$H shifts at the diastereotopic C-6 position, all PPAPs investigated here (**3**–**5**) have the C-7 *endo* configuration (Supplementary Table 2). These findings further support the theory that C-7 *endo* stereochemistry is important for the assembly of caged scaffolds, as spatial proximity to C-3 may be required for intramolecular cyclization.

## Conspicuous hybrid aspartate-rich motif suggests target sequences

A transcriptomic database was constructed de novo using *H. sampsonii* RNA. Based on the working hypothesis that bifunctional PTs, which catalyse both prenylation and cyclization, may be the target catalysts, *Hs*PT8px served as a probe to identify homologous amino acid sequences. *Hs*PT8px and *Hs*PTpat are the only aromatic PTs characterised from *H. sampsonii*[28] (NCBI accession numbers AZK16224.1 and AZK16225.1, respectively). They catalyse the successive *gem*-diprenylation of tetrahydroxyxanthone, which is a direct descendant of the phlorbenzophenone framework (Supplementary Fig. 14). Phylogenetic analysis of the 13 transcriptomically selected homologues suggested a xanthone-specific clade that accommodated *Hs*PT8px but did not help uncover putative bifunctional PT candidates (Supplementary Fig. 15). However, closer examination of the two conserved aspartate-rich loop regions responsible for the coordination of metal ion cofactors was able to guide the selection of two PT candidate sequences (Fig. 3). These conserved motifs are present in all plant PTs and known to comprise distinct canonical sequences in aromatic PTs of the homogentisate (HG) and *p*-hydroxybenzoate (PHB) families[29]. Interestingly, all homologues assembled in this study shared the expected HG-specific motifs, with the exception of candidates 2948c0g1 and 2948c0g2. These two sequences had unique hybrid motifs, which included a PHB-like sequence in loop 2 and retained the HG consensus in loop 6, making them stand out from previously characterised PTs. Their shared amino acid sequence identity was 64.7% at 96.2% coverage (69.7% identity at 100% coverage outside the first 100 residues including the plastidial signal peptide), while the nearest putative PT homologue shared 51.4% and 51.0% sequence identity with 2948c0g1 and 2948c0g2, respectively.

## *Hs*CPTa and *Hs*CPTb catalyse prenylative cyclizations to yield type A and B PPAPs

The two cDNA sequences 2948c0g1 and 2948c0g2 were cloned into the episomal vector pESC-URA, which enables gene expression in yeast (*Saccharomyces cerevisiae*) under the control of the inducible GAL1 promoter. Microsomal fractions containing the recombinant proteins were incubated in PT assays. Both enzymes catalysed prenylative cyclization, using **1** as acceptor substrate and dimethylallyl pyrophosphate (DMAPP) as donor substrate (Fig. 2b, c). Interestingly, the regiospecificities of the cyclization reactions differed. The type A product **3** was formed by the translation product of transcript 2948c0g1 and the enzyme was named *Hs*CPTa (Fig. 2b). In contrast, the type B product **5** was formed by the translation product of transcript 2948c0g2 and the enzyme was named *Hs*CPTb (Fig. 2c). *Hs*CPTa was also able to accept substrate **2**, resulting in the formation of **4** as another type A product. Both enzymes failed to accept geranyl pyrophosphate (GPP) as the donor substrate. Control assays containing microsomes from empty vector-transformed yeast cells lacked enzyme activity (Fig. 2d). Furthermore, assays with phlorbenzophenone confirmed that *Hs*CPTa and *Hs*CPTb, unlike other aromatic PTs, failed to catalyse the transfer of a prenyl group to the aromatic core, underlining their specific role in the catalysis of alkene-intercepted prenylation. All enzymatic products were identified by comparing their

to require the formation of activated epoxide intermediates[24,26]. The resulting pyran-fused or adamantyl-like consolidations are structural elements of many elucidated hypersampsone[26] and sampsonione type[6,24] molecules (Supplementary Fig. 13). In vitro propagated and pot-cultivated *H. sampsonii* plants used in this study contained high levels of **3** and **4**, which are direct precursors of the adamantane PPAPs plukenetione A and otogirinin A, respectively. This accumulation is presumably due to the downregulation of the yet unidentified late pathway steps leading to caged PPAPs. Compounds **3** and **4** are supposed to have a high turnover in the plant's metabolism, given the high number of isolated compounds postulated as their putative downstream products. In contrast, **5** has been commonly identified in *H. sampsonii* extracts[6], which might be related to the comparatively low diversity of reported type B derivatives.

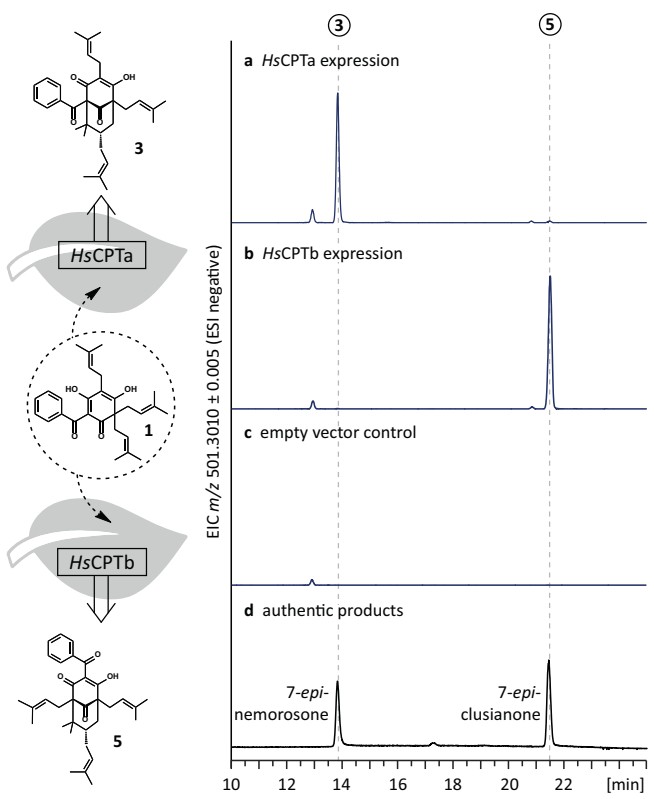

**Fig. 4 | Type A and B product formation in _N. benthamiana_ leaves.** The _Hs_CPTa and _Hs_CPTb genes were transiently expressed in _N. benthamiana_ leaves, and transgenic disk sections were incubated with the acceptor substrate grandone **1**. **a**, **b** Extracted ion chromatograms (_m/z_ [M–H]⁻ = 501.3010 ± 0.005) confirmed the formation of 7-_epi_-nemorosone **3** and 7-_epi_-clusianone **5** in leaves expressing _Hs_CPTa and _Hs_CPTb, respectively. **c** Neither product was observed in leaves infiltrated with the GFP-encoding control construct. **d** Authentic reference compounds.

retention times, UV spectra, and tandem-MS fragmentation patterns with those of the structure-elucidated constituents of _H. sampsonii_ leaves (Supplementary Fig. 16).

These findings highlight that plants are able to regulate the biosynthesis of type A and B molecules by employing highly specialised enzymes. Their kinetic characterisation revealed similar parameters for the common acceptor substrate, grandone **1**, which were comparable to the data reported for the patulone forming enzymes, _Hs_PT8px and _Hs_PTpat, used above as probes. _Hs_CPTa preferentially accepted grandone **1** with a $K_M$ matching that of DMAPP (62.01 ± 10.06 μM and 65.44 ± 6.63 μM, respectively). The affinity to the secondary acceptor substrate, kolanone **2**, was lower (114.24 ± 20.03 μM; Supplementary Fig. 17). _Hs_CPTb exclusively accepted grandone **1** with an affinity comparable to that of _Hs_CPTa (40.36 ± 5.06 μM). It had a relatively low affinity towards DMAPP (297.79 ± 29.49 μM), which is still in the range reported for aromatic PTs[30]. While optimising the assay parameters, the preferred pH values were found to differ. While _Hs_CPTa functioned best at pH 7, _Hs_CPTb reached the highest activity at pH 10 (Supplementary Fig. 17). This prompted us to investigate whether the enzymes might be present in different subcellular compartments in planta. However, localisation of translational reporter fusions in transiently transformed _Nicotiana benthamiana_ leaves provided evidence that both enzymes are associated with the chloroplast envelope (Supplementary Fig. 18). Although the outer or inner membrane of the chloroplast envelope may be differentially targeted by the enzymes, the observed discrepancy in the pH optima is likely to be an artifact resulting from the in vitro reconstruction of the enzymatic reactions. In fact, there is evidence that some aromatic PTs are involved in heteromeric enzyme complexes[31],

potentially adding to other effects of their specific membranous microenvironment.

To confirm the detected activities of _Hs_CPTa and _Hs_CPTb in the plant environment, we incubated disk sections of transiently transformed _N. benthamiana_ leaves in the presence of acceptor substrate **1**. Consistent with the results of the in vitro assays, the type A product **3** was exclusively detected in leaves expressing _Hs_CPTa (Fig. 4a), whereas the type B product **5** was solely formed in samples expressing _Hs_CPTb (Fig. 4b). No enzymatic activity was observed in leaves infiltrated with a GFP-encoding control construct (Fig. 4c). Moreover, no significant differences in overall product yields were detected between leaves expressing _Hs_CPTa or _Hs_CPTb genes, as confirmed by three biological repeats.

**Inverted substrate binding modes result in type A and B PPAPs**

In molecular docking simulations on refined in silico structures generated by AlphaFold2, the binding cavities predicted for _Hs_CPTa and _Hs_CPTb were too small to accommodate the extended geranyl chain of acceptor substrate **2** to generate docked conformations that could effectively explain the experimental results. To overcome this problem, we referenced the crystal structures of homologous UbiA-type PTs (PDB ID: 4tq3–6, 6m31, 6m34, 7BPU, 5OON). While differences in the core structure of the crystals (transmembrane helices αII–VII) are minimal, the relative positions of αI (N-terminal) and αVIII–IX (C-terminal) adapt to changes in substrate size. Thus, to allow the docking of **2** with minimal adjustments, we modelled the N- and C-terminal helices after 4tq3, which contains a hydrophobic tunnel for the putative binding of polyprenyl chains (Supplementary Fig. 19)[32]. We constructed the binding of Mg²⁺ ions and DMAPP based on the highly conserved aspartate-rich motifs of 4tq3 (Supplementary Fig. 20). According to the known initiation of prenyl transfer reactions, the dimethylallyl cation was dissociated from the pyrophosphate moiety, setting the C-O distance to 2.8 Å, which is common for such reaction transition states[33,34]. In addition, electron densities of the activated intermediate obtained from quantum mechanical calculations at the B3LYP-D3/(6-31 G + (d)) level in the gas phase indicated that a nucleophilic attack of C-1 and C-5 on the prenyl cation requires the deprotonation of a hydroxy group (C-4 or C-6). Consequently, we constrained putative hydrogen bonds between the hydroxy groups and potentially assisting deprotonation residues in the subsequent induced-fit docking procedure.

The binding modes of the acceptor substrate **1**, which fit best the chemical logic of the reactions, specified binding energies of −105.76 kcal mol⁻¹ and −85.48 kcal mol⁻¹ in _Hs_CPTa and _Hs_CPTb, respectively. The binding poses of **1** rotate around the C-3 to C-6 axis of the phloroglucinol ring in the two enzymes, presenting opposing prenyl residues of the C-3 _gem_-diprenyl group towards the prenyl donor (Fig. 5 a, b). The _trans_-like conformation of the benzoyl moiety at C-1 and the prenyl residue at C-5 is characteristic of both binding modes. It ensures the regiospecificity of the cyclization reaction by restricting the access to the carbon with the upward-facing substituent. The OH group at C-6 of the acceptor substrate forms a hydrogen bond with the N153 residue of _Hs_CPTa and the H142 residue of _Hs_CPTb to facilitate the deprotonation. Furthermore, the specific binding mode in _Hs_CPTa appears to be stabilised by π-π interactions between the benzoyl group of the substrate and the F111 side chain of the protein. In _Hs_CPTb, the benzoyl group may turn towards the internal hydrophobic region of the binding cavity due to the polarity difference of the C100 residue.

Docking of substrate **2** into _Hs_CPTa suggests a similar binding mode and reaction principle (Fig. 5c). The elongated C-3 geranyl chain does not affect the reaction mechanism, as it is buried in the hydrophobic part of the binding tunnel and thus faces away from the prenyl donor. However, in the inverted binding mode of _Hs_CPTb (Fig. 5d), the bulky geranyl chain interferes with the binding of the donor substrate

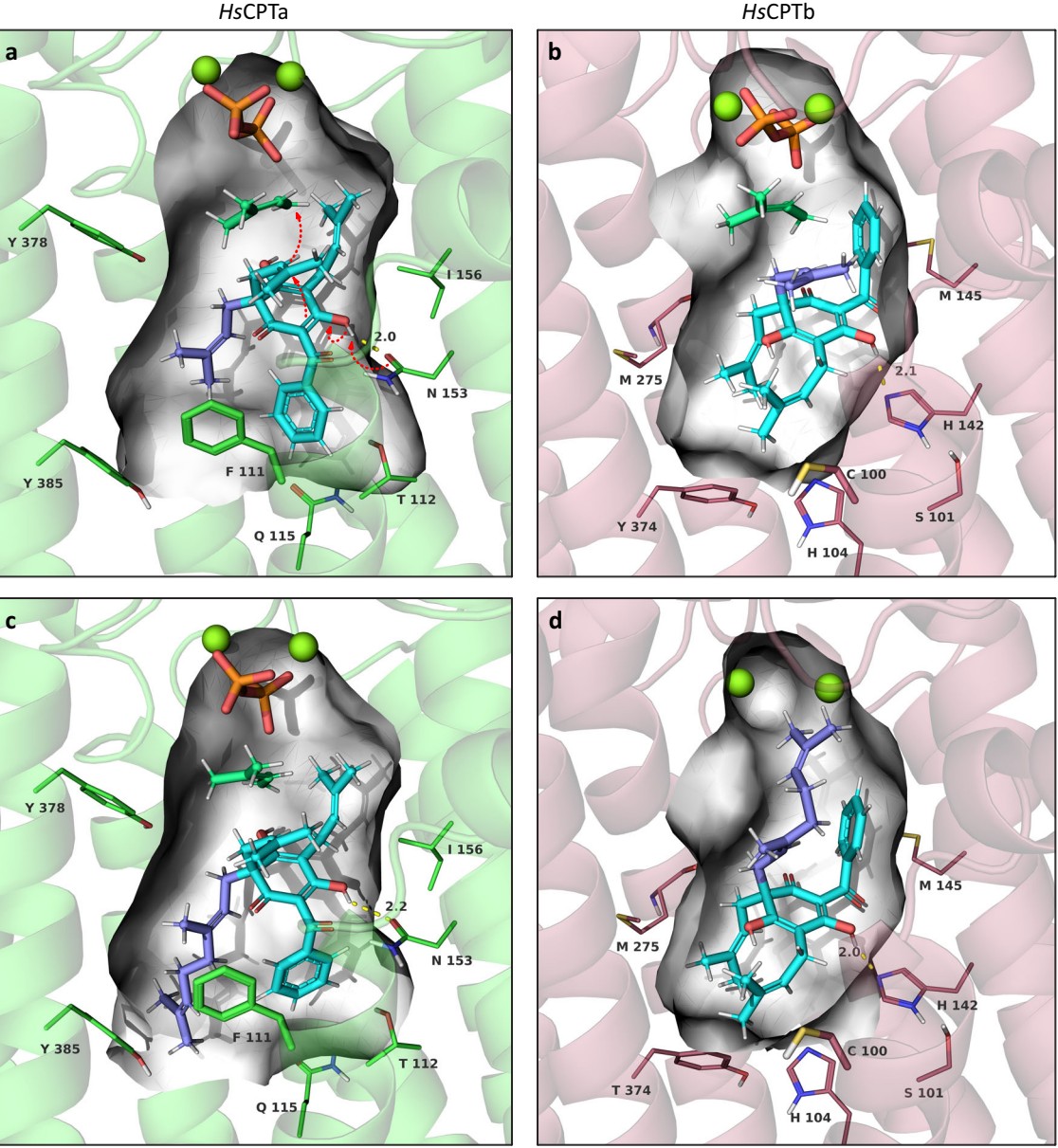

**Fig. 5 | Inverted acceptor substrate binding modes of *Hs*CPTa and *Hs*CPTb.** Simulation included the aspartate-mediated coordination of $Mg^{2+}$ ions (spheres), the interacting pyrophosphate moiety (orange), and the dissociated isoprenyl residue (turquoise) (for details of the conserved donor substrate binding, see Supplementary Fig. 20). **a, b** Inverted binding poses of acceptor substrate **1** in the active site cavities of *Hs*CPTa (green) and *Hs*CPTb (red). The two C-3 *gem*-prenyl residues involved in the reactions are distinguished by light and deep blue colours. Steric hindrance is introduced by the *trans*-like conformation of the benzoyl (C-1) and prenyl groups (C-5) with respect to the phloroglucinol ring plane, leading to the regiospecific ring closure reactions. **c** Docking of acceptor substrate **2** into the *Hs*CPTa pocket allows the same reaction principle, which is not affected by the elongated geranyl group (deep blue). **d** In the inverted binding mode of *Hs*CPTb, the upward-facing geranyl chain interferes with the binding of the donor substrate, preventing the reaction from proceeding. White surface, binding cavity; dashed red arrows, reaction sequence; dashed yellow lines, distance measure in Å.

and no reaction proceeds. This simulation is in good agreement with the experimental results, given that **2** was only converted by *Hs*CPTa.

The outlined binding modes are supported by the previously established relationships between the absolute configurations and the specific optical rotations of type A and B PPAPs[19,35–37]. Depending on which of the two C-3 *gem*-prenyl residues of the acceptor substrate is activated by prenyl transfer, the subsequent cyclization yields different enantiomers by bridging opposite sides of the phloroglucinol ring (Supplementary Fig. 21). Crucially, type A and B ring closures on opposite faces of the ring yield products with equal optical activities. As the relative configuration of (+)-**3** can only be inferred from the type A framework of a synthetic (−)-nemorosone precursor[19], studies of the Porco group, in combination with their earlier work on the synthesis

of (−)-clusianone[36], were critical to support the described relationships. The specific rotations measured for the *H. sampsonii* PPAPs **3, 4** and **5** were $[\alpha]_D$ = +82.4, +110.3 and +46.7, respectively. Thus, the binding orientations of the acceptor substrate in *Hs*CPTa and *Hs*CPTb must be inverted to form the specific product rotamers.

### Specific amino acid residues control the regiospecificities of the prenylative cyclizations

Mutational studies were designed to identify key amino acid residues that confer regiospecificity to the reactions. In a first holistic approach, we gradually transformed *Hs*CPTb into *Hs*CPTa by C-terminal exchange of incrementally larger fragments (Fig. 6a and Supplementary Fig. 22). Interestingly, up to 192 residues could be replaced

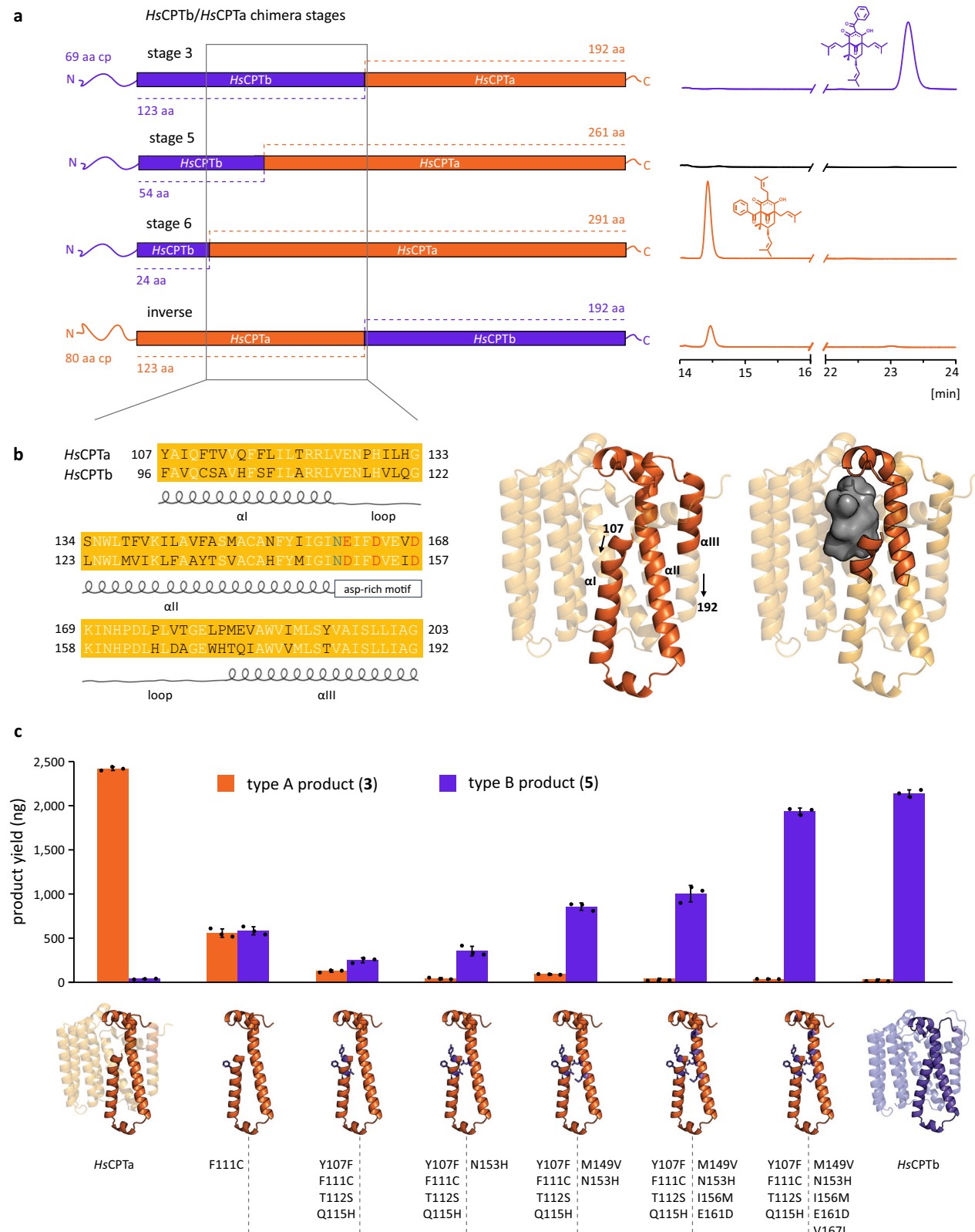

**Fig. 6 | Mutagenic transformation of *Hs*CPTa into *Hs*CPTb. a** Progressive C-terminal exchange of *Hs*CPTb residues with corresponding fragments of *Hs*CPTa identified a 96 amino acids region essential for regiospecificity (box). **b** Sequence alignment of this region of interest, its location in the predicted *Hs*CPTa fold (opaque model), and its interface to the central cavity (grey surface). **c** Progressive conversion of the *Hs*CPTa regiospecificity into the *Hs*CPTb regiospecificity using expanding sets of amino acid substitutions. Total amounts of type A and B products obtained in single standardised reactions are indicated. Data are means ± SD of three biological replicates.

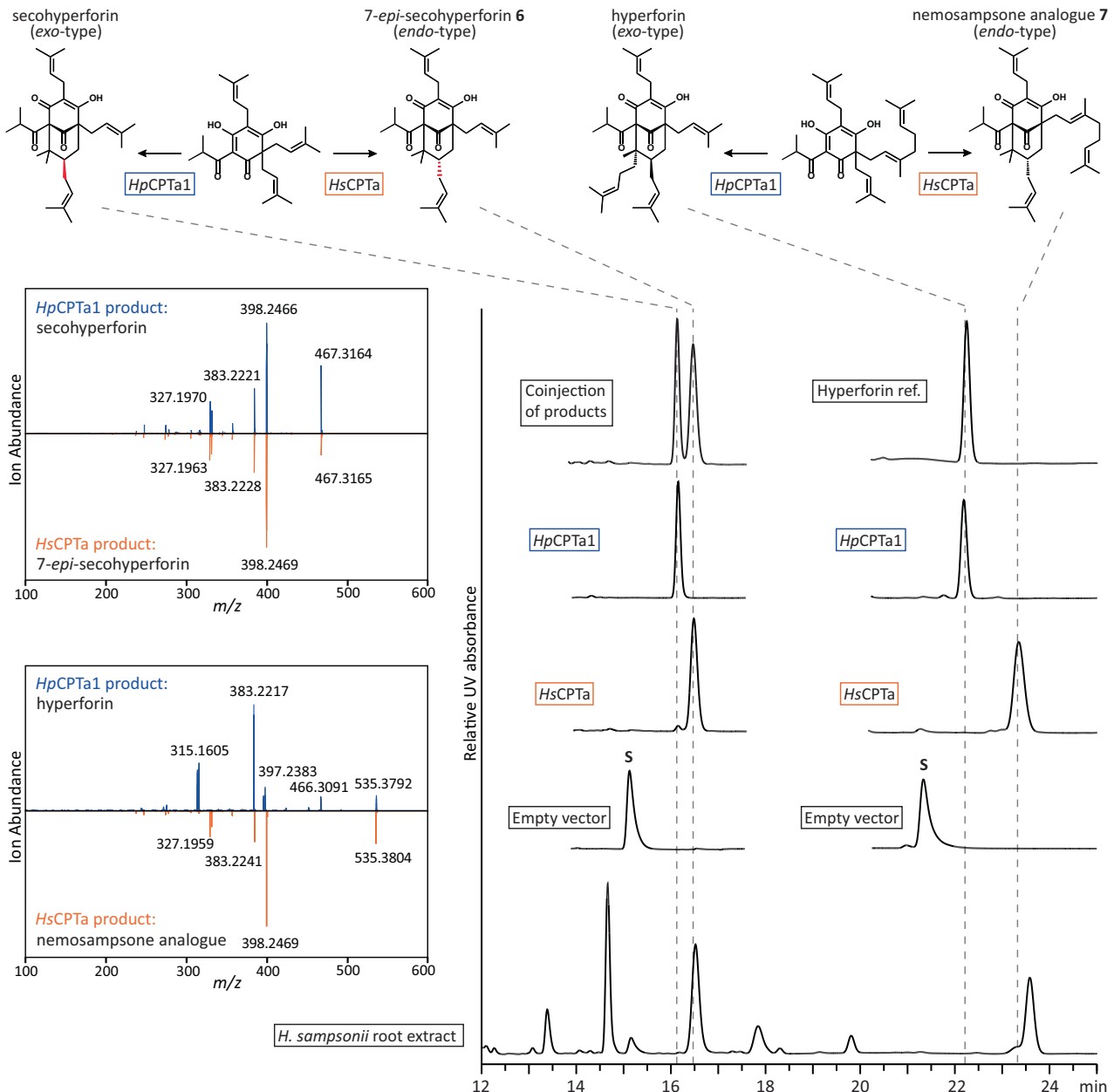

**Fig. 7 | Divergent activities of *Hs*CPTa and *Hp*CPTa1 yield hyperforin analogues.** HPLC-DAD analysis of enzyme assays containing recombinant yeast microsomes. *Hs*CPTa and *Hp*CPTa1 catalysed the conversion of triprenylated phlorisobutyr-ophenone to secohyperforin *endo* and *exo* epimers, respectively. *Hs*CPTa transformed diprenylated-monogeranylated phlorisobutyrophenone to the aliphatic nemosampsone analogue **7**, whereas *Hp*CPTa1 catalysed a different mode of cyclization that resulted in the formation of hyperforin. No conversions were observed in control assays containing microsomes from empty vector-transformed yeast cells. S, substrate peak.

without affecting catalytic efficiency and regiospecificity (chimera stages 1–3). Further substitutions rendered the chimeric *Hs*CPTb inactive (stages 4–5). The substitution of 291 C-terminal residues gave rise to a chimera with restored functionality and switched regiospecificity (stage 6). Notably, the stage 3 inversed chimera confirmed that 123 N-terminal residues of both *Hs*CPTb and *Hs*CPTa are sufficient for the selective formation of their respective regiomers. These experiments highlighted a stretch of 96 amino acids as the primary region of interest. This was further substantiated by modelling and docking simulations with reciprocal *Hs*CPTa and *Hs*CPTb chimeras comprising the exchanged 96 residues region, which predicted inverted substrate binding modes similar to the docked conformations in their wild-type counterparts (Supplementary Fig. 23).

Subsequent site-directed mutagenesis focused on the divergent residues in this region, particularly those directly associated with the modelled binding pocket (Fig. 6b). An overview of all tested mutants is provided in Supplementary Fig. 24. The most insightful combinations of amino acid substitutions, which underpin the complete transformation of *Hs*CPTa as a type A cyclase into a type B cyclase (Fig. 6c), are as follows. A single point mutation, F111C, located at the beginning of the first transmembrane helical segment (αI) completely abolished the regiospecificity of *Hs*CPTa. While the incorporation of three additional mutations Y107F, T112S and Q115H (αI) slightly shifted the conversion in favour of the type B product, introduction of N153H into the second transmembrane helix (αII) solidified this change in selectivity. The overall diminished activity of the mutant was partially restored by

including the additional point mutation M149V (αII). To ultimately produce a mutant activity equivalent to *Hs*CPTb, the incorporation of three further mutations, I156M, E161D and V167I was found to be essential. The latter two were located at the first aspartate-rich motif of the enzyme.

Of the nine mutations identified to control the catalytic transformation of *Hs*CPTa, seven (excluding M149V and V167I) were predicted to interact directly with the substrate binding pocket. While each of those seven substitutions cooperatively contributed to the stabilisation of the type B specific binding pose, the most dramatic effect was observed for mutations in the F111/C100 position. As hypothesised based on the obtained docking results, the F111 residue appears to be critical for ensuring type A regiospecificity, most likely by interacting directly with the C-1 benzoyl moiety of the acceptor substrate. In the F111C point mutant, this missing interaction seems to enable both binding orientations at the cost of diminished activity. The same observation could not be made in the completely inactivated *Hs*CPTb C100F mutant. It appears that the bulky phenyl ring restricts the volume of the binding pocket, as the relatively small size of C100 would normally allow the C-5 prenyl chain of the substrate to protrude beyond this residue in the wild-type enzyme (Fig. 5b, d). Although not in direct proximity to the binding pocket, the two remaining mutations, M149V and V167I, are likely to influence the helix spacing between αI and αII or helix loops HL23 and HL67, respectively. Rather than affecting the regiospecificity, these substitutions appear to influence the overall performance of the enzyme.

### *Hs*CPTa and its *H. perforatum* homologue *Hp*CPTa1 catalyse the formation of hyperforin analogues

PPAPs with aliphatic acyl groups, such as hyperforin and its derivatives, are renowned constituents of *Hypericum perforatum* (St. John's wort, Supplementary Fig. 25), but uncommon in *H. sampsonii*. Metabolic profiling of *H. sampsonii* root, stem, and leaf extracts revealed the presence of varying levels of phlorbenzophenone derivatives in all tissues (Supplementary Fig. 26a, b). Interestingly, the root extract additionally contained two compounds, **6** and **7**, with mass and UV signatures consistent with the well-studied phlorisobutyrophenone-derived PPAPs secohyperforin and hyperforin, respectively. However, comparison with available reference compounds revealed different retention times. Isolation and structure elucidation of compound **6** yielded NMR data very similar to those reported for the type A PPAP secohyperforin[38] (Supplementary Table 3 and Supplementary Fig. 27). Crucially, the chemical shifts that differed from the reference values were associated with the critical positions C-6, C-7, and C-8. They were consistent with the values expected for *endo*-PPAPs (Supplementary Table 2). Thus, the structure of **6** was established as 7-*epi*-secohyperforin. In contrast, compound **7** that shared the same molecular mass as hyperforin was not unambiguously elucidated in this study due to its instability and limited amount. However, its MS/MS data resembled the pattern produced by nemosampsone **4** and clearly differed from that of hyperforin (Supplementary Fig. 28). Therefore, its structure is postulated to be the aliphatic nemosampsone analogue **7**. Given the absence of further transcriptomic CPT candidates, we speculated that *Hs*CPTa may be responsible for the formation of PPAPs with an aliphatic acyl group in the roots of *H. sampsonii*. This hypothesis was supported by RT-qPCR results, revealing a higher relative expression of *Hs*CPTa than *Hs*CPTb in the roots (Supplementary Fig. 26c).

To identify *Hs*CPTa homologues from *H. perforatum* potentially involved in the final step of the biosynthesis of hyperforins (Supplementary Fig. 25), the *Hs*CPTa sequence was used as a query to search the Medicinal Plant Genomics Resource database (MPGR, mpgr.uga.edu). The first BLAST hit (hpa_locus_470_iso_1_len_1384_ver_2), named *Hp*CPTa1, shared 64.5% amino acid sequence identity with *Hs*CPTa. It also contained the same variation in the metal ion binding

motifs. Interestingly, the transcripts of *Hp*CPTa1 were upregulated in flower tissues, as evidenced by the FPKM values (Supplementary Table 4), aligning with the expected expression profile of hyperforin biosynthetic genes. Notably, hyperforin accumulation in flowers and fruits of *H. perforatum* is well-established[39].

In vitro assays with recombinant yeast microsomes confirmed that both *Hs*CPTa and *Hp*CPTa1 accepted the prenyl donor DMAPP to convert the triprenylated phlorisobutyrophenone substrate, colupulone, into distinct products (Fig. 7). Two secohyperforin epimers were formed, which eluted less than 30 seconds apart and produced identical MS/MS fragments. In agreement with the isolated metabolites, the *H. sampsonii* enzyme catalysed the formation of 7-*epi*-secohyperforin (*endo* configuration), whereas the *H. perforatum* enzyme yielded secohyperforin (*exo* configuration). Both enzymes also accepted the geranyl-bearing precursor and catalysed reactions with different cyclization modes. In analogy to the conversion of acceptor substrate **2**, *Hs*CPTa directed the initial prenyl transfer to the C-3 prenyl group of the precursor, whereas *Hp*CPTa1 prenylated the C-3 geranyl group to form hyperforin. No activity was detected in *Hs*CPTb-containing assays and control assays with microsomes from empty-vector transformed cells.

These results demonstrate that *H. sampsonii* roots are another source of phlorisobutyrophenone-derived PPAPs, which are structurally different from the metabolites found in *H. perforatum*. To the best of our knowledge, no example of a type B PPAP with an aliphatic acyl group has been reported in the literature. This finding is consistent with the inability of *Hs*CPTb to act on the corresponding substrate. The kinetic characterisation of colupulone in *Hs*CPTa-catalysed reactions revealed parameters similar to those of substrate **2** ($K_M$, 105.55 ± 9.86 μM; Supplementary Fig. 17), reconfirming grandone **1** as the preferred acceptor substrate.

In conclusion, we present the discovery of two enzymatic blueprints, which confirm the biosynthetic scenario of prenylative cyclization in *H. sampsonii*, and characterise their sophisticated mechanisms, which enable *H. sampsonii* to regulate the regiodivergent formation of diverse type A and B PPAP compounds. The sequence information obtained, especially for the aspartate-rich regions, will help elucidate the biosynthetic machinery that yields other valuable PPAPs in the Hypericaceae and Clusiaceae families, as exemplified here by detection of the (seco)hyperforin-forming enzyme in *H. perforatum*. These discoveries are anticipated to contribute to deciphering further biosynthetic steps via phylogenetic and co-expression analyses. Gaining access to these biosynthetic gateway reactions is paramount to the development of synthetic biology approaches aiming at the production of medicinal PPAPs. The differences in the metabolomes of in vitro propagated and field-grown *H. sampsonii* plants need confirmation by more extensive analysis but may provide a rare opportunity to study the biosynthesis of caged PPAPs with adamantane and homoadamantane scaffolds via comparative transcriptomics.

## Methods
### Plant material
In vitro-propagated *Hypericum sampsonii* Hance (Hypericaceae) plants were established at the Institute of Botany, the Chinese Academy of Sciences, Beijing, China[40]. Seeds were surface sterilised and germinated on MS medium (Duchefa Biochemie, Haarlem, The Netherlands) supplemented with 2% sucrose and 0.7% agar. Seedlings were cultured on the same medium containing 0.5% activated charcoal to keep roots in the dark and absorb harmful root exudates. For propagation, shoots were cut into segments with at least one axillary bud forming plantlets on the same medium within two months. Following one-week-acclimatisation, rooted plantlets were transferred to pots containing commercial soil for garden plants and cultivated in a whole-plant culture room (Institute of Pharmaceutical Biology, Braunschweig) at 22 °C, 70% relative humidity, and 16/8 (light/dark) photoperiod.

## Extraction and isolation of prenylated benzophenone derivatives

Leaves of four-month-old plants (10 g dry weight) were lyophilised and homogenised (M20 Universal mill; IKA, Staufen, Germany). Eight aliquots of the resulting powder were transferred to 50 mL conical reaction tubes and extracted three times with 50 mL dichloromethane. The combined extract was centrifuged, the supernatant filtrated through filter paper, and the solvent evaporated under $N_2$ gas. The residue (1 g) was fractionated using an Isolera One Flash Purification Chromatography system (Biotage, Uppsala, Sweden), equipped with a Chromabond Flash BT cartridge (SiOH, 40–63 μm, 40 g; Macherey-Nagel, Düren, Germany). Separation was stretched over 23 column volumes (CV), using n-hexane (A) and ethyl acetate (B) as mobile phase (gradient: 2 CV 0% B, 6 CV 0–10% B, 15 CV 10–60% B). A total of 30 fractions a 45 mL were collected, automatically controlled by detection of UV signals in the λ-all mode. After solvent evaporation under $N_2$ gas, the fractions were passed through a Chromabond HR-X SPE column (6 mL, 55–60 μm; Macherey-Nagel) to remove chlorophyll. Lipophilic target compounds were isolated by semi-preparative HPLC on a normal-phase EC NUCLEODUR column (100-5 silica, 4.6×250 mm, 5 μm; Macherey-Nagel) with a flow rate of 3 mL min⁻¹. The mobile phase consisting of n-hexane (A) and ethyl acetate (B) was used for isocratic elution at 3% B for 16 min. Separation of fractions 16 and 17 yielded pure nemosampsone **4** (7 mg), while fraction 18 yielded 7-*epi*-nemorosone **3** (6 mg), compound **4** (4 mg), and 7-*epi*-clusianone **5** (3 mg). More polar target compounds were purified on a reversed-phase EC NUCLEODUR column (π², 10 × 250 mm, 5 μm; Macherey-Nagel) at a flow rate of 3 mL min⁻¹ using 0.1% formic acid in water (A) and acetonitrile (B) as mobile phase (80–92% B from 0–15 min). Separation of fractions 23–28 resulted in the isolation of grandone **1** (5 mg) and kolanone **2** (2 mg). 7-*epi*-secohyperforin **6** (0.28 mg) was isolated on the above normal-phase HPLC system from fresh *H. sampsonii* roots used for RNA isolation.

## NMR spectroscopy

Measurements were carried out on either a Bruker Advance III HD 500 MHz NMR spectrometer (Bruker Biospin, Karlsruhe, Germany) or a Bruker Advance III HD 700 MHz spectrometer, both equipped with cryoplatforms and TCI cryoprobes. The 1D and 2D spectral data were obtained from ¹H NMR, ¹³C NMR, DEPTQ, ¹H-¹H COSY, ¹H-¹H ROESY, ¹H-¹³C HSQC, and ¹H-¹³C HMBC experiments. Spectrometer control and data processing were accomplished using the Bruker TopSpin software v3.6.1. Standard Bruker pulse programs were used for NMR data acquisition.

## Polarimetry experiments

Specific optical rotations were measured on a Jasco P-2000 polarimeter (Jasco, Gross-Umstadt, Germany), equipped with a cylindrical quartz cell (3.5 mm diameter, 100 mm optical path length). Measurements were carried out at 20 °C using sodium light at 589 nm, setting the source aperture to 3 mm. Sample concentrations (w/v %) of 7-*epi*-nemorosone **3**, nemosampsone **4**, and 7-*epi*-clusianone **5** were 0.22, 0.20, and 0.06, respectively.

## RNA isolation and sequencing

Shoots and roots of two-month-old *H. sampsonii* plants were homogenised in liquid nitrogen using mortar and pestle. Total RNA was extracted from the fine powders (-100 mg) using the InviTrap Spin Plant RNA mini kit (Invitek Molecular, Berlin, Germany). After removal of residual genomic DNA by digestion with DNAse I (Thermo Fisher, Waltham, USA), RNA was purified using the Monarch RNA Cleanup kit (New England Biolabs, Ipswich, USA). Quality and quantity of total RNA were determined on a Bioanalyzer 2100, paired with the RNA 6000 Nano LabChip kit (both Agilent, Santa Clara, USA). Samples with RIN > 7.0 were accepted for further work-up. Library preparation, RNA-sequencing, and transcriptome assembly were performed at LC Sciences (Houston, USA). Poly(A) mRNA was isolated from total RNA (~1 μg) using oligo(dT)-coupled magnetic beads (Thermo Fisher) and fragmented into small pieces using divalent cations at elevated temperature. RNA fragments were reverse-transcribed to prepare the final strand-specific cDNA library according to the dUTP method[41]. The average insert size for the paired-end libraries was 300 ± 50 bp. Paired-end 2 × 150 bp sequencing was performed on an Illumina Hiseq 4000 platform (Illumina, San Diego, USA), following the vendor recommended protocol. 54 to 57 million RNA-Seq reads were collected per sample.

## De novo transcriptome assembly, annotation, and evaluation

Trimmomatic[42] software was used for adaptor trimming and removal of reads containing low quality or undetermined bases. Quality control was performed by FastQC[43]. The clean data of ensured quality was used for downstream analysis. De novo assembly of the transcriptome was performed using the Trinity[44] method. The longest transcript in each Trinity cluster was defined as the unigene. All assembled unigenes were functionally annotated by alignment against non-redundant (NR) NCBI protein database, Gene ontology (GO), SwissProt, Kyoto Encyclopedia of Genes and Genomes (KEGG), and EggNOG, using DIAMOND[45] software with default parameters ($e \leq 1^{-5}$). A translated nucleotide BLAST search (tblastn) identified homologous sequences, employing the amino acid sequence of *Hs*PT8px as query (NCBI accession number AZK16224.1). Amino acid sequence alignment utilised the MUSCLE[46] algorithm. Phylogenetic analysis relied on MEGA11[47] software using the maximum-likelihood method (JTT + F G5)[48], as recommended by the in-built evaluation tool. The quality of the phylogenetic tree was inferred for each node by computation of 500 bootstrap replicates.

## Molecular biology essentials

Our molecular cloning toolbox comprised an assortment of commercial products and services. Synthesis of modified and unmodified primers and sequencing of DNA samples were performed by Eurofins Genomics (Ebersberg, Germany) and Microsynth Seqlab (Göttingen, Germany), respectively. PCR reactions were conducted using either PCRBIO HiFi polymerase (PCR Biosystems, London, UK) or PfuTurbo Cx HotStart DNA polymerase (Agilent), if encountering uracil stalling. Clean-up of DNA fragments from agarose gels or PCR reactions was done using the innuPREP DOUBLEpure kit (Analytik Jena, Jena, Germany). Restriction enzymes for cloning and control digestions and T4 DNA ligase for ligation of blunt or sticky ends were purchased from Thermo Fisher. Plasmid DNA was isolated using the QIAprep™ Spin Miniprep kit (Qiagen, Hilden, Germany). Unless indicated otherwise, all reactions using commercial products were performed according to the manufacturers' guidelines.

## Gene expression analysis by RT-qPCR

Total *H. sampsonii* root, stem, and leaf RNA (1 μg) served for cDNA synthesis using the iScript Reverse Transcription Supermix for RT-qPCR (Bio-Rad, Hercules, USA). RT-qPCR reactions were set up using the iTaq Universal SYBR Green Supermix (Bio-Rad) according to the manufacturer's guidelines and run on a CFX Connect Real-Time PCR Detection System (Bio-Rad). The 2-step PCR method comprised an initial denaturation step at 95 °C for 2 min, followed by 40 cycles at 95 °C for 5 s and 60 °C for 30 s, recording plate data after each annealing/extension step. Melting curves ensuring the amplification specificity were recorded after each run from 65 to 95 °C in 0.5 °C increments over 5 s. Efficiencies of primers designed for the quantification of *Hs*CPTa, *Hs*CPTb and the housekeeping genes ACT2 and TUB-B (Supplementary Table 5) were tested against a serial dilution made with pooled cDNA of all tissues. Data analysis was performed using the Bio-Rad CFX Manager v3.1 software (Bio-Rad).

## Functional expression in *Saccharomyces cerevisiae*

The binary pESC-URA vector allowed inducible gene expression in *S. cerevisiae* and efficient cloning and plasmid propagation in *Escherichia coli*. Construct assembly was achieved by a standard restriction-ligation technique, utilising MCS2 for the insertion of target genes under control of the yeast GAL1 promoter. Inserts were amplified using overhang primers introducing the desired restriction sites (Supplementary Table 5). Both backbone and insert were digested with *Bam* HI and *Kpn* I prior to assembly. Reaction products were directly used for transformation of *E. coli* DH5α by the common $CaCl_2$ heat-shock method and transformants were recovered on LB agar plates (0.5% yeast extract, 1% peptone, 1% NaCl) supplemented with $100\,\mu g\,mL^{-1}$ ampicillin. After ensuring correct assembly by sequencing, expression constructs were introduced into *S. cerevisiae* INVsc1 (Thermo Fisher), using the Frozen-EZ Yeast Transformation II kit (Zymo Research Europe, Freiburg, Germany). *S. cerevisiae* clones were selected on SGI plates (0.67% yeast nitrogen base, supplemented with amino acids, 2% glucose) lacking uracil as an auxotrophic marker. For protein production, single yeast colonies were suspended in 5 mL uracil-deficient SGI medium (100 mL flasks). After 24 h incubation at 30 °C and 200 rpm, around 2 mL were used to start 150 mL main cultures (1 L flasks, $OD_{600} = 0.01$) in YPEG medium (1% yeast extract, 1% peptone, 3% ethanol, 0.5% glucose). After incubation for 28–30 h, heterologous gene expression was induced by adding 2% galactose and cultures were incubated for another 16 h to accumulate heterologous enzymes. Cells were harvested by centrifugation at 2500 x *g* for 5 min and rinsed twice with 10 and 5 mL TEK buffer (50 mM Tris-HCl pH 7.4, 1 mM EDTA, 100 mM KCl), prior to mechanical disruption by vigorous vortexing (30 min, 4 °C) with glass beads (3 g, 0.45 mm) in 3 mL TES buffer (50 mM Tris-HCl, pH 7.4, 1 mM EDTA, 600 mM sorbitol). To preserve enzyme stability, all following steps were conducted at 4 °C. After centrifugation at 2500 x *g* for 5 min, the protein-containing supernatant was collected and the remaining cell debris extracted two more times with 3 ml TES buffer. Membrane fractions were pelleted by ultra-centrifugation of the combined extract at 100,000 x *g* for 1 h and the pellet was resuspended in 2 mL buffer A (50 mM Tris-HCl pH 7.4, 300 mM NaCl, 10% glycerol).

## Enzyme assays and characterisation

Standard reactions for activity screening (125 μL) contained 100 μg microsomal protein, 5 mM $MgCl_2$, 400 μM DMAPP, 200 μM acceptor substrate, buffered by 100 mM Tris-HCl pH 8. DMAPP was synthesised as previously described[49]. Assays were incubated at 37 °C for 1 h at constant agitation, stopped with 17.5 μL 2 N HCl and extracted with 500 μL ethyl acetate. The solvent was evaporated *in vacuo*, the residue taken up in 100 μL methanol, and 50 μL analysed by HPLC. For enzyme characterisation, assay parameters were individually investigated: temperature (25–50 °C), pH (6–11), protein concentration (10–500 μg per assay), and reaction time (2–200 min). Under optimum conditions for each enzyme, the $K_M$ values were determined at DMAPP saturation, while eight concentrations of **1** and **2** ranged from 0.1 to 3 times $K_M$. Maximum velocities were deduced from Michaelis Menten kinetics in analogy to $V_{max}$. Curve-fitting was managed using OriginPro2022 (OriginLab, Northampton, USA) with the provided Enzyme Kinetics plugin. Means and standard deviations were calculated from three technical replicates, for which exact amounts of microsomal protein from the same batch were provided. The product yield of all *Hs*CPTa and *Hs*CPTb mutants created was determined by single standardised reactions using 200 μg microsomal protein to convert 15 pmol donor and acceptor substrates in Tris-HCl pH 8.5 buffer for 45 min at 40 °C. To account for differences in the relative protein amounts obtained from yeast cultivation and induction procedure, reactions were repeated three times using membrane fractions from three individual cultivations.

## Transient gene expression in *Nicotiana benthamiana*

Assembly of pCambia 2300 u 35 S expression constructs was achieved by adapting the USER™ strategy[50]. One-pot reactions were set up using uracil-containing PCR products and vectors in a 10:1 ratio. Restriction enzyme *Pac* I, nicking enzyme *Nt.Bbv* CI (NEB), and USER enzyme (NEB) worked simultaneously in rCutSmart buffer (NEB). One-pot excision and hybridisation were facilitated at alternating temperatures (20 min at 37 °C, 15 cycles of 5 min at 37 °C and 2 min at 25 °C, 20 min at 25 °C). Sequence-confirmed constructs were transferred into *Agrobacterium tumefaciens* EHA105 cells, using the established freeze-thaw technique. After selection of positive transformants, pre-cultures were inoculated with a single colony in 10 mL LB medium (50 μg mL⁻¹ kanamycin, 50 μg mL⁻¹ rifampicin) and incubated overnight at 28 °C and 220 rpm. On the next day, 5 mL were used to start 50 mL main-cultures (100 mL in case of P19 helper-strain), which were incubated for an additional 24 h. Agrobacteria were pelleted by centrifugation at 2000 x *g* for 10 min, briefly washed with water and diluted to an $OD_{600}$ of 0.5 with infection medium (10 mM MES, 10 mM $MgSO_4$, 100 μM acetosyringone pH 5.5). Virulence was induced for 2 h at 25 °C and constant agitation. All tested strains were mixed 1:1 with a helper-strain expressing the commonly used P19 inhibitor of gene silencing[51]. 4–5-week-old *N. benthamiana* plants were infected by syringe-infiltration into the abaxial side of the two youngest fully expanded leaves. Plants were left to dry in the dark and subsequently transferred to the above-mentioned whole-plant culture room.

## *Nicotiana benthamiana* leaf disk experiments

To confirm the activities of *Hs*CPTa and *Hs*CPTb in planta, 1 cm diameter disk sections of transiently transformed *N. benthamiana* leaves were collected 4 days after *Agrobacterium* infiltration. Based on the previously described protocol[52], the leaf sections were transferred to a 48-well plate containing 250 μL Tris-HCl (100 mM, pH 7) and 20 μM acceptor substrate. The plate was sealed with parafilm to prevent buffer evaporation and incubated at plant growth conditions (22 °C, 70% relative humidity, 16/8 light/dark cycles). After 48 h, leaf discs were placed in 2 mL screw cap tubes, flash-frozen in liquid nitrogen and homogenised using a MM400 Tissuelyser (Retsch, Haan, Germany) and 0.7 mm garnet beads. Metabolites were extracted with 500 μL ethyl acetate and cell debris was removed by repeated centrifugation at 19,000 x *g* for 10 min. After solvent evaporation under $N_2$ gas, residues were taken up in 100 μL MeOH for LC-MS analysis.

## Analytical procedures

HPLC analysis was performed on a VWR Hitachi LaChrom Elite machine, equipped with a L-2130 pump, L-2200 autosampler, and L-2455 DAD detector. Samples were run on a ZORBAX Eclipse Plus column (C18, 4.6 × 100 mm, 3.5 μm; Agilent) at a flow rate of 1 mL min⁻¹, using 0.1% formic acid in water (A) and acetonitrile (B) as mobile phase (gradient: 50–70% B from 0–5 min, 70–92% B from 5–10 min, 92–96% B from 10–28 min). For HRESI-MS characterisation, the HPLC system was coupled to a Bruker Compact QToF mass spectrometer (Bruker Daltonics, Bremen, Germany), operated via the Compass HyStar control platform (Bruker Daltonics). Measurements were performed in negative ionisation mode at 30,000 m Δm⁻¹ resolving power, recording ions in the full scan range of *m/z* 50–1300. The *m/z* values obtained were recalibrated using the expected cluster ion *m/z* values of a sodium formate-isopropanol solution, which was injected at the beginning of each run. For LC-MS/MS analysis, the multiple reaction monitoring (MRM) mode of the instrument was used, setting the collision energy to 30, 35, and 40 eV.

## Enzyme engineering

Stepwise substitution of the *Hs*CPTb C-terminus with corresponding amino acids of *Hs*CPTa was performed using the one-pot fusion PCR method[53]. *Hs*CPTa fragments designated for substitution were

amplified and purified from agarose gel. In subsequent fusion PCR, they functioned as megaprimers for the initial amplification of chimerical sequences, using gene-specific pESC-URA forward primers and *Hs*CPTb as the template. After 5 cycles, pESC-URA reverse primers were added to increase the fidelity of the reaction and introduce the required restriction sites. To eliminate any residual template DNA, PCR products were digested with *Dpn* I (2–3 h). The resulting hybrid genes were cloned into pESC-URA, as described above. Site-directed mutagenesis served to introduce point mutations to *Hs*CPTa and *Hs*CPTb constructs by full replication of the plasmid, using tailored pairs of partially overlapping complementary primers (Supplementary Table 5), as previously described[54]. After removal of parental DNA by *Dpn* I digestion and transfer to *E. coli*, the nicked plasmids were repaired by virtue of the bacterial DNA maintenance machinery.

## Homology modelling and substrate docking

The initial structures of *Hs*CPTa and *Hs*CPTb were modelled using AlphaFold2 (DeepMind, London, England) and further refined with Prime (Schrödinger, New York, USA), utilising implicit membrane and OPLS4 force fields. The resulting structures were validated using the SAVES6 server (saves.mbi.ucla.edu). Protein-ligand interactions were investigated using Schrödinger's suite of molecular modelling tools (Schrödinger Release 2018-2) by employing the induced-fit docking (IFD) approach, which accounts for ligand and protein flexibility during docking simulations. To accommodate the substrate in the smaller binding pocket during the docking process, both the substrate and the ligand van der Waals radii scaling factor were set to 0.3. Taking into account the size of the ligand, the docking box size was set to 15 Å with the docking centre defined by the centroid of the C-alpha atoms of four residues that envelop the binding site. After docking, clustering of the results was performed and a representative conformation from each cluster was selected as the starting structure for further docking refinement, using the extra precision (XP) mode. Upon completion of the docking, we assessed the binding free energies of each binding conformation using the generalised Born surface area (MM/GBSA) method[55] and selected the most favourable binding conformation. The LigPrep module (Schrödinger) was used to generate ligand 3D structures with optimised geometries and appropriate ionisation states at the preferred pH values of 7 and 10 for computations involving *Hs*CPTa and *Hs*CPTb, respectively. The representative membrane topology of *H. sampsonii* PTs was visualised based on transcript 8613c0g2 using MembraneFold[56].

## Subcellular localisation

Two versions of the original binary plant-expression vector pCambia 2300 u 35 S were utilised, designated for translational fusion to either N- or C-terminus of YFP reporter. Four days after *N. benthamiana* leaf infiltration, localisation of YFP-fused enzymes was determined by confocal laser scanning microscopy. Simple preparations of abaxial epidermal cells were mounted in water and analysed on the confocal laser scanning microscope cLSM 880 (Zeiss, Oberkochen, Germany), equipped with a Plan-Apochromat 20x/0.8 air objective (Zeiss). Excitation with a 488 nm argon laser allowed comparison of chlorophyll autofluorescence to the emissions of YFP-fused localisation markers. Results observed were confirmed in three biological replicates.

## Reporting summary

Further information on research design is available in the Nature Portfolio Reporting Summary linked to this article.

## Data availability

The nucleotide sequences of *Hs*CPTa, *Hs*CPTb, and *Hp*CPTa1 are deposited in the National Center for Biotechnology (NCBI) GenBank under the accession numbers OR464113, OR464114, and PP538028. respectively. The raw reads obtained from the RNA sequencing of

*H. sampsonii* root and shoot samples are deposited in the NCBI Sequence Read Archive (SRA) under the BioProject accession PRJNA1010338. Source data are provided with this paper.

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

## Acknowledgements

The authors thank Prof. Paraskev T. Nedialkov of the Medical University of Sofia, Bulgaria for graciously providing the 7-*epi*-clusianone reference compound, Dr. Veit Grabe of the Max Planck Institute for Chemical Ecology, Jena, Germany for his kind assistance in operating the confocal laser scanning microscope, and Dr. Till Beuerle, TU Braunschweig, for kindly handling the tandem mass spectrometry infusions. This research was supported by the Deutsche Forschungsgemeinschaft (DFG, German Research Foundation, LI 3400/4-1), the National Key Research and Development Program of China (2018YFA0900300), and Tianjin Synthetic Biotechnology Innovation Capacity Improvement Projects (TSBI-CIP-PTJS-002). H.M.B.S. acknowledges support from the German Egyptian Research Long-Term Scholarship (GERLS) programme funded by the Egyptian Ministry of Higher Education and the Deutscher Akademischer Austauschdienst (DAAD). Open access funding enabled and organised by Project DEAL.

## Author contributions

L.E., L.B., I.E-A., and B.L. devised the experimental design. L.E. conducted cloning, expression, and characterisation of biosynthetic genes in *S. cerevisiae* and *N. benthamiana*, generated reciprocal mutants, and performed activity assays. H.L. extracted and purified metabolites from *H. sampsonii* and performed NMR experiments. H.L. and C.P. interpreted NMR data. P.L. performed molecular modelling and docking simulations and analysed the results. H.M.B.S. and I.E-A. carried out synthesis and purification of donor substrates. H.M.B.S. and T.M. identified and assayed *Hp*CPTa1. H.M. provided advice and assisted with bioinformatic analyses. L.E., L.B., and B.L. wrote the manuscript. All authors read and approved the manuscript.

## Funding

## Competing interests

The authors declare no competing interests.
