## [Peer Review File · Nature Communications]

Regiodivergent biosynthesis of bridged bicyclononanesReviewer #1 (Remarks to the Author):

The manuscript presented the identification of four known and one new type A or type B PPAPs from *Hypericum sampsonii* plants, as well as the isolation of two enzymes that regiodivergently convert a common precursor into type A and B products. The underlying mechanism for regulating the regiodivergent formation of type A and B PPAPs was explored through molecular modeling and docking studies. The enzymatic activities were interconverted using reciprocal mutagenesis. The work firstly elucidate the unique biochemistry that yields type A and B bicyclo[3.3.1]nonane cores in plants; although the precise mechanisms were not fully elucidated, and it could be acceptable after the authors address the following issues.

In 'HsCPTa and HsCPTb catalyse prenylative cyclizations to yield type A and B PPAPs' section:

'All enzymatic products were identified by comparing their retention times, UV spectra, and tandem-MS fragmentation patterns with those of the structure-elucidated constituents of *H. sampsonii* leaves (Supplementary Fig. 15).'

- 1, The high-resolution mass spectrometry data is essential to substantiate the structural identification of the products, especially considering that they were not prepared separately.
- 2, How about the acceptor selectivity of HsCPTa and HsCPTb? Can they catalyze the prenylation on phlorbenzophenone core?
- 3, The KM values for DMAPP in HsCPTa and HsCPTb reactions were suggested to be provided.

In 'Inverted substrate binding modes result in type A and B PPAPs' section:

'... the binding cavities predicted for HsCPTa and HsCPTb were too small to accommodate the extended geranyl chain of acceptor substrate 2. ... Thus, the N- and C-terminal helices of HsCPTa (α I and α IX, Supplementary Fig. 17) were adjusted to allow the docking of 2.'

- 1, The mutagenesis results have demonstrated that substitutions of certain residues, even those not in proximity to the binding pocket, can still exert an influence on the reaction. Therefore, could the author provide a rational explanation for manual adjustment of the predicted structures?
- 2, The setting of the binding site, including the grid box and its size, is crucial for obtaining accurate docking results. However, these specific details were not provided in this article.

In 'Specific amino acid residues control the regiospecificities of the prenylative cyclizations' section:

- 1, What are the activities of chimeras with exchange in the '96 amino acids' region between HsCPTb and HsCPTa?
- 2, Can the presence of inverted substrate bindings be observed in the structures of chimeras and mutants exhibiting switched regiospecificity?

Reviewer #2 (Remarks to the Author):

The manuscript by Ernst et al. describes the discovery and biochemical characterization of two bifunctional aromatic prenyltransferases from the plant *Hypericum sampsonii*, responsible for both prenylation and cyclization reactions in the biosynthesis of hyperforin analogs. The authors investigated their function both in vitro using yeast microsomes and in vivo in tobacco through transient expression and feeding assays. They purified both the substrates and products of these prenyltransferases from the plant source and conducted convincing NMR/UV/MS studies on these chemicals.

Despite sharing the same substrate, the two PTs varied in their cyclization mode, producing either A or B type products. The authors further explored the amino acid residues responsible for this preference through domain swapping, modeling, docking, and site-directed mutagenesis. They identified several key residues in two α -helices responsible for the distinct binding/cyclizing modes. The experiments were well-executed, and the data were well-presented and convincing. This discovery not only represents a key reaction for bicyclo[3.3.1]nonane analogs in *H. sampsonii* but also likely contributes to the biosynthesis of a more well known metabolite, hyperforin in *H. perforatum*, St. John's wort.

Thus, an immediate question arising from this work is whether a homolog or homologs of

HsCPTa/b are found in *H. perforatum*, which would use an aliphatic instead of an aromatic intermediate for hyperforin biosynthesis. These reactions are essentially the same but involve slightly different substrates and cyclization mode. If so, do they contain similar amino acids in the two helices, which give respective cyclization modes? Perhaps there are residue differences accommodating the variations in methyl/phenyl substitution. Will recombinant *H. perforatum* enzyme(s) catalyze reactions using *H. sampsonii* intermediates and vice versa? Addressing these questions will significantly improve the impact of the current manuscript. Both the genome and multiple transcriptomes are available for *H. perforatum*, and gene synthesis is low cost and convenient. I strongly recommend that the authors conduct these experiments.

Some minor points:

1. The amino acid identity between HsCPTa and CPTb, and between them and other putative PTs should be mentioned in the text to provide perspective on these two enzymes.
2. In the supplementary information, the authors mentioned that transcriptomes were generated from multiple tissues. It would be useful to present chromatograms of prenylated phloroglucinol metabolites and transcript levels of both CPTs and other PTs, and to discuss the relationship between transcription levels and metabolite levels in various tissues. It is also unclear why these two candidates were selected other than the small difference in the metal-binding region.
3. Are HsCPTa/b the only two PTs in the plant that showed the variation in metal-binding region? Any other homologs may be identified in the plants?
3. Please include a supplementary figure showing the reactions of HsPT8px and HsPTpat for readers to understand their roles and relationships to the two PTs.
4. Does *H. sampsonii* contain hyperforin or other analogs derived from isobutyric acid instead of benzoic acid?
5. A discussion on the structural difference between hyperforin and the metabolites in this study is welcome.

Point-by-point response to the reviewers' comments

Title: Regiodivergent biosynthesis of bridged bicyclononanes

Dear Editor, dear Reviewers,

Thank you so much for the detailed and constructive criticism, which led us to substantially improve the quality of our manuscript. We did our best to address all concerns raised as completely as possible in our point-by-point response below. All changes made are marked in yellow throughout the revised manuscript. To comply with the Nature Communications publishing format, all Extended Data Figures and Tables were moved to the Supplementary Information and the numbering was changed accordingly.

Reviewer #1:

The manuscript presented the identification of four known and one new type A or type B PPAPs from Hypericum sampsonii plants, as well as the isolation of two enzymes that regiodivergently convert a common precursor into type A and B products. The underlying mechanism for regulating the regiodivergent formation of type A and B PPAPs was explored through molecular modeling and docking studies. The enzymatic activities were interconverted using reciprocal mutagenesis. The work firstly elucidate the unique biochemistry that yields type A and B bicyclo[3.3.1]nonane cores in plants; although the precise mechanisms were not fully elucidated, and it could be acceptable after the authors address the following issues.

In 'HsCPTa and HsCPTb catalyse prenylative cyclizations to yield type A and B PPAPs' section:

'All enzymatic products were identified by comparing their retention times, UV spectra, and tandem-MS fragmentation patterns with those of the structure-elucidated constituents of H. sampsonii leaves (Supplementary Fig. 15).'

1. *The high-resolution mass spectrometry data is essential to substantiate the structural identification of the products, especially considering that they were not prepared separately.*

Response: We completely agree with the reviewer that the high-resolution MS data should be provided to ensure the identity of all enzymatic products. We repeated all tandem-MS experiments using a capable instrument and include the high-resolution data in the revised Supplementary Fig. 16. The information about the mass spectrometer and measurement was updated in the Methods section of the revised manuscript as follows.

“For HRESI-MS characterization, the HPLC system was coupled to a Bruker Compact QToF mass spectrometer (Bruker Daltonics, Bremen, Germany), operated via the Compass HyStar control platform (Bruker Daltonics). Measurements were performed in negative ionization mode at 30,000 m Δm^{-1} resolving power, recording ions in the full scan range of m/z 50–1300. The m/z values obtained were recalibrated using the expected cluster ion m/z values of a sodium formate-isopropanol solution, which

was injected at the beginning of each run. For LC-MS/MS analysis, the multiple reaction monitoring (MRM) mode of the instrument was used, setting the collision energy to 30, 35, and 40 eV.”

2. *How about the acceptor selectivity of HsCPTa and HsCPTb? Can they catalyze the prenylation on phlorbenzophenone core?*

Response: In our *in vitro* assays, both HsCPTa and HsCPTb were unable to accept the unsubstituted benzoylphloroglucinol core as a substrate. Indeed, the catalysed alkene-intercepted prenyl transfer reactions differ from the activity of known plant aromatic prenyltransferases and resemble the activity of ‘head-to-middle’ acyclic terpene synthases. We thank the reviewer for highlighting this important point and note this finding in the revised manuscript as follows.

“Furthermore, assays with phlorbenzophenone confirmed that HsCPTa and HsCPTb, unlike other aromatic PTs, failed to catalyse the transfer of a prenyl group to the aromatic core, underlining their specific role in the catalysis of alkene-intercepted prenylation.”

3. *The K_M values for DMAPP in HsCPTa and HsCPTb reactions were suggested to be provided.*

Response: Thank you for bringing this up. We determined the K_M values of DMAPP in the HsCPTa and HsCPTb reactions. In response to the remarks of reviewer #2, we also expanded the search of possible substrates and provide the kinetic parameters for colupulone as another acceptor substrate in HsCPTa reactions. We revised the relevant parts of the manuscript as follows.

“Their kinetic characterization revealed similar parameters for the common acceptor substrate, grandone **1**, which were comparable to the data reported for the patulone forming enzymes, HsPT8px and HsPTpat, used above as probes. HsCPTa preferentially accepted grandone **1** with a K_M matching that of DMAPP (62.01 ± 10.06 μM and 65.44 ± 6.63 μM, respectively). The affinity to the secondary acceptor substrate, kolanone **2**, was lower (114.24 ± 20.03 μM; Supplementary Fig. 17b). HsCPTb exclusively accepted grandone **1** with an affinity comparable to that of HsCPTa (40.36 ± 5.06 μM). It had a relatively low affinity towards DMAPP (297.79 ± 29.49 μM), which is still in the range reported for aromatic PTs³⁰.”

“The kinetic characterization of colupulone in HsCPTa-catalysed reactions revealed parameters similar to those of substrate **2** (K_M, 105.55 ± 9.86 μM; Supplementary Fig. 17), reconfirming grandone **1** as the preferred acceptor substrate.”

In ‘Inverted substrate binding modes result in type A and B PPAPs’ section:

‘... the binding cavities predicted for HsCPTa and HsCPTb were too small to accommodate the extended geranyl chain of acceptor substrate **2**. ... Thus, the N- and C-terminal helices of HsCPTa (α I and α IX, Supplementary Fig. 17) were adjusted to allow the docking of **2**.’

1. The mutagenesis results have demonstrated that substitutions of certain residues, even those not in proximity to the binding pocket, can still exert an influence on the reaction. Therefore, could the author provide a rational explanation for manual adjustment of the predicted structures?

Response: We deeply appreciate the reviewer's request for a more detailed explanation of the structural adjustments that we made to the predicted protein structure. Upon obtaining and refining the initial structures predicted by AlphaFold2, we attempted substrate docking. Several docked conformations were generated but none of them could effectively explain the experimental phenomena. The structural differences between the two products detected in the experiments led us to consider the possibility of inverted substrate binding modes. Such a possibility would require a larger binding cavity, especially for accommodating the geranyl-bearing substrate kolanone. Therefore, we compared homologous proteins that could be searched in the protein structure database (4tq3–6, 6m31, 6m34, 7BPU, 5OON) and examined the sizes of their substrates. We observed that the differences in these crystal structures are minimal in the transmembrane region α II–VII, whereas the relative positions of α I (N-terminal) and α VIII–IX (C-terminal) indeed adjust to the changes in substrate size. Notably, adjustments of α I and α IX are most pronounced. This is as if the α II–VII region forms the palm of a hand which is relatively stable in structure, while α I and α VIII–IX form thumb and fingers, playing a role in grasping the reactant. Since we wanted to achieve substrate accommodation with minimal adjustments, we focused on tweaking α I and α IX in this study. In the end, this was realised by referencing the helix positions of 4tq3 which was described to comprise a hydrophobic substrate tunnel that likely accommodates the binding of polyprenyl chains (Huang *et al.*, 2014; 10.1371/journal.pbio.1001911). Regarding reciprocal mutagenesis, we may note that the two essential mutations, M149V and V167I, are not located in proximity to the binding pocket. These mutations affected the overall product yields but had little effect on regiospecificity. The pivotal determinant of product selectivity was the mutation F111/C100 as described in the manuscript. We revised the initial paragraph of the relevant chapter as follows.

“To overcome this problem, we referenced the crystal structures of homologous UbiA-type PTs (PDB ID: 4tq3–6, 6m31, 6m34, 7BPU, 5OON). While differences in the core structure of the crystals (transmembrane helices α II–VII) are minimal, the relative positions of α I (N-terminal) and α VIII–IX (C-terminal) adapt to changes in substrate size. Thus, to allow the docking of **2** with minimal adjustments, we modelled the N- and C-terminal helices after 4tq3, which contains a hydrophobic tunnel for the putative binding of polyprenyl chains (Supplementary Fig. 19)³²”

2. The setting of the binding site, including the grid box and its size, is crucial for obtaining accurate docking results. However, these specific details were not provided in this article.

Response: We are very grateful to the reviewer for identifying gaps in our description of the docking parameters. Our initial manuscript indeed did not provide sufficient details on this aspect which is crucial for the reproducibility and understanding of our docking approach. In response to the reviewer's

comment, we revised the Methods section to include the following comprehensive description of the docking parameters used in this study.

“Protein-ligand interactions were investigated using Schrödinger's suite of molecular modelling tools (Schrödinger Release 2018-2) by employing the induced-fit docking (IFD) approach, which accounts for ligand and protein flexibility during docking simulations. To accommodate the substrate in the smaller binding pocket during the docking process, both the substrate and the ligand van der Waals radii scaling factor were set to 0.3. Taking into account the size of the ligand, the docking box size was set to 15 Å with the docking center defined by the centroid of the C-alpha atoms of four residues that envelop the binding site. After docking, clustering of the results was performed and a representative conformation from each cluster was selected as the starting structure for further docking refinement, using the extra precision (XP) mode. Upon completion of the docking, we assessed the binding free energies of each binding conformation using the generalized Born surface area (MM/GBSA) method⁵⁵ and selected the most favorable binding conformation.”

In ‘Specific amino acid residues control the regiospecificities of the prenylative cyclizations’ section:

1. *What are the activities of chimeras with exchange in the ‘96 amino acids’ region between HsCPTb and HsCPTa?*

Response: Thank you for raising this question. The activities of all HsCPTb/HsCPTa chimeras generated in this study are shown in Supplementary Fig. 22 (former Extended Data Fig. 2). The chimeras were constructed as an initial approach to identify potential domains involved in the observed regiospecificities. The findings led us to conclude that the preservation of a ‘96 amino acids region’ is essential for the enzymes to maintain their respective activities. Two chimeras (stages 4 and 5) with disrupted/mixed domains were tested and exhibited impaired enzymatic activities. All further investigations of the highlighted region focused on the analysis of point mutations and their combinations, which were selected on the basis of their likely interaction with the predicted binding pocket. The complete activity data of all tested mutants is illustrated in Supplementary Fig. 23 (former Extended Data Fig. 3), which was carefully revised to improve readability.

2. *Can the presence of inverted substrate bindings be observed in the structures of chimeras and mutants exhibiting switched regiospecificity?*

Response: Thank you for this interesting question. Indeed, our modelling and docking simulations did not include the chimeric enzymes with switched regiospecificity. We agree with the reviewer that doing so is a way to cross-validate the observed binding modes. Therefore, we modelled the structures of reciprocal HsCPTa and HsCPTb chimeras, in which the regiospecificity-controlling regions were exchanged, and performed substrate docking. The simulation results confirm that inverted binding modes that rationalize the switched regiospecificities can be observed in the two mutants (Supplementary Fig. 24). Since the AF2 models of the two chimeras are based on templates for protein

structure prediction, the structural differences are mainly reflected in the changes of the amino acid residues in the exchanged region. As depicted in the new Supplementary Fig. 24, the docked substrate binding mode in the *HsCPTa(b)* chimera was similar to that of *HsCPTb* and vice versa. This result seems reasonable when considering that the majority of residues at the interface of the binding pocket outside the exchanged regiospecificity-determining region are conserved between the two enzymes. The following statement is included in the revised manuscript.

“This was further substantiated by modelling and docking simulations with reciprocal *HsCPTa* and *HsCPTb* chimeras comprising the exchanged 96 residues region, which predicted inverted substrate binding modes similar to the docked conformations in their wildtype counterparts (Supplementary Fig. 24).”

Reviewer #2:

*The manuscript by Ernst et al. describes the discovery and biochemical characterization of two bifunctional aromatic prenyltransferases from the plant *Hypericum sampsonii*, responsible for both prenylation and cyclization reactions in the biosynthesis of hyperforin analogs. The authors investigated their function both in vitro using yeast microsomes and in vivo in tobacco through transient expression and feeding assays. They purified both the substrates and products of these prenyltransferases from the plant source and conducted convincing NMR/UV/MS studies on these chemicals.*

*Despite sharing the same substrate, the two PTs varied in their cyclization mode, producing either A or B type products. The authors further explored the amino acid residues responsible for this preference through domain swapping, modeling, docking, and site-directed mutagenesis. They identified several key residues in two alpha-helices responsible for the distinct binding/cyclizing modes. The experiments were well-executed, and the data were well-presented and convincing. This discovery not only represents a key reaction for bicyclo[3.3.1]nonane analogs in *H. sampsonii* but also likely contributes to the biosynthesis of a more well known metabolite, hyperforin in *H. perforatum*, St. John's wort.*

*Thus, an immediate question arising from this work is whether a homolog or homologs of HsCPTa/b are found in *H. perforatum*, which would use an aliphatic instead of an aromatic intermediate for hyperforin biosynthesis. These reactions are essentially the same but involve slightly different substrates and cyclization mode. If so, do they contain similar amino acids in the two helices, which give respective cyclization modes? Perhaps there are residue differences accommodating the variations in methyl/phenyl substitution. Will recombinant *H. perforatum* enzyme(s) catalyze reactions using *H. sampsonii* intermediates and vice versa? Addressing these questions will significantly improve the impact of the current manuscript. Both the genome and multiple transcriptomes are available for *H. perforatum*, and gene synthesis is low cost and convenient. I strongly recommend that the authors conduct these experiments.*

Response: Thank you very much for raising this interesting issue. Of course, we share the reviewer's interest in hyperforin as the best-known PPAP and the underlying biosynthetic process in *H. perforatum*. Initially, we decided to put the focus of the present manuscript on the regiodivergent catalysis of type A and type B prenylative cyclizations, exemplified by the formation of 7-*epi*-nemorosone and 7-*epi*-clusianone, respectively, which are major constituents of *H. sampsonii*. Hyperforin is an additional type A compound but absent from *H. sampsonii*. Instead, it is present in *H. perforatum* which, however, lacks type B PPAPs. For these reasons, we planned a separate investigation and publication of hyperforin biosynthesis.

However, also taking into account the reviewer's points #2 and #5, we found that the constituents of the much more complex root extract of *H. sampsonii* include mass signatures of PPAPs with aliphatic acyl groups, which did not match any of our hyperforin-type reference compounds. Encouraged by the reviewer, we therefore decided to conduct a series of additional experiments to investigate the

biosynthesis of yet unknown aliphatic acyl residue-bearing PPAPs in *H. sampsonii* roots and compare it to the biosynthesis of hyperforins in *H. perforatum*.

The results are presented in a new and separate section of the revised manuscript, entitled HsCPTa and its *H. perforatum* homolog HpCPTa1 catalyse the formation of hyperforin analogues. It includes the identification of the enzyme responsible for the formation of hyperforin and secohyperforin in *H. perforatum*. The enzyme shares 64.5% amino acid sequence identity with HsCPTa and contains the same variation in the metal ion binding motifs as described for HsCPTa and HsCPTb. As the reviewer points out, the type A cyclization mode is slightly different in the *H. perforatum* enzyme (named HpCPTa1), which attaches the final prenyl group to the geranyl side chain. Furthermore, it forms *exo* type PPAPs, whereas all *H. sampsonii* products have *endo* configuration. By testing the aliphatic precursors as substrates for HsCPTa and HsCPTb, we characterised HsCPTa to be responsible for the formation of phlorisobutyrophenone-derived PPAPs in *H. sampsonii* roots. As depicted in the new Fig. 7 in the main text, HsCPTa and HpCPTa1 convert the triprenylated substrate into distinct products, which elute less than 30 seconds apart and exhibit identical MS/MS fragmentation spectra. We managed to purify enough HsCPTa product from the root extract to be able to elucidate its structure as 7-*epi*-secohyperforin, which is the *endo* diastereomer of secohyperforin. In contrast to the triprenylated substrate, the geranyl-bearing hyperforin precursor was converted to a product that showed a different MS/MS spectrum and eluted over one minute apart. Due to the low abundance of this compound in *H. sampsonii* roots, we cannot provide conclusive structural data for this constituent, but based on the MS/MS data we are confident that it is the aliphatic analogue of nemosampsonone.

Consistent with the inability of HsCPTa to catalyse the formation of hyperforin, HpCPTa1 does not produce nemosampsonone. Instead, HpCPTa1 accepts kolanone to form putative bzhyperforin, which has recently been identified in *Hypericum forrestii* (Lu *et al.*, 2020; 10.1039/D0QO00152J) but not yet been reported for *H. perforatum*. While we acknowledge the significance of HpCPTa1 as a key enzyme for the formation of hyperforin, its detailed biochemical and mechanistic characterization is beyond the scope of the present manuscript, especially because preliminary results indicate that a complicated biosynthetic network underlies hyperforin formation in *H. perforatum*.

Some minor points:

1. *The amino acid identity between HsCPTa and CPTb, and between them and other putative PTs should be mentioned in the text to provide perspective on these two enzymes.*

Response: We completely agree with the reviewer that amino acid sequence identities should be provided to the reader. This information was added to the manuscript as follows.

“Their shared amino acid sequence identity was 64.7% at 96.2% coverage (69.7% identity at 100% coverage outside the first 100 residues including the plastidial signal peptide), while the nearest putative PT homolog shared 51.4% and 51.0% sequence identity with 2948c0g1 and 2948c0g2, respectively.”

2. In the supplementary information, the authors mentioned that transcriptomes were generated from multiple tissues. It would be useful to present chromatograms of prenylated phloroglucinol metabolites and transcript levels of both CPTs and other PTs, and to discuss the relationship between transcription levels and metabolite levels in various tissues.

Response: Thank you for bringing this up. We analysed extracts of the available root, stem, and leaf material and depict the metabolic profiles in the new Supplementary Fig. 26a. The same material was used to isolate total RNA for RT-qPCR analysis. Quantification of metabolite levels and relative expression data are shown in Supplementary Fig. 26b and c, respectively. These experiments showed that *HsCPTa* and *HsCPTb* products and transcripts are present throughout the analysed organs. The levels of transcripts and metabolites largely correlated. The relatively high expression of *HsCPTa* in the roots can be explained by its additional responsibility in the synthesis of aliphatic root derivatives. The relatively low *HsCPTa* transcript level in leaves may be due to the advanced age of the leaves studied. Meanwhile, *HsCPTb* is most highly expressed in the stem, which is consistent with the high content of 7-*epi*-clusianone in the stem extract.

It is also unclear why these two candidates were selected other than the small difference in the metal-binding region.

3. Are *HsCPTa/b* the only two PTs in the plant that showed the variation in metal-binding region? Any other homologs may be identified in the plants?

Response: We fully understand the reviewer's reservation about the significance of the variations in the metal ion binding motifs. We are also hesitant to speculate about their implications. Although a further example from *H. perforatum* has now been identified (*HpCPTa1*), the elucidation of further CPTs is necessary to clarify whether this motif in fact is a recurrent feature of enzymes with cyclase functionality. Nevertheless, the motif was an important part of our candidate selection process. No other sequence motif specific of CPTs was detected aside the general PT sequence variability, making it difficult to select CPT candidates based on phylogenetic analysis. Notably, we did not find any other CPT variant or transcript with divergent residues in this region among all assembled homologs. For this high conservation, these motifs were previously exploited by us to amplify PT genes without access to transcriptomic information (Nagia *et al.*, 2018; 10.1111/nph.15611 and Fiesel *et al.*, 2015; 10.3390/molecules200915616).

4. Please include a supplementary figure showing the reactions of *HsPT8px* and *HsPTpat* for readers to understand their roles and relationships to the two PTs.

Response: Thank you for this good suggestion. We included this information in the additional Supplementary Fig. 14.

5. Does *H. sampsonii* contain hyperforin or other analogs derived from isobutyric acid instead of benzoic acid?

Response: Thank you for raising this point. Our metabolite analysis of different plant organs indicates that compounds derived from isobutyric acid appear to be exclusively present in the roots. The two isobutyric acid-derived PPAPs that were identified in root extracts are 7-*epi*-secohyperforin and a putative nemosampsonone analogue. As we had access to only a limited number of plant organs in this study, we cannot rule out the possibility that such compounds may also be present in buds, flowers or seeds. It will be of interest to readers that the model compound hyperforin was not detected in any of the analysed *H. sampsonii* extracts. Nor could we confirm the presence of any type B compounds derived from isobutyric acid in any parts of the plant, which is in agreement with the observed activity of HsCPTb.

6. A discussion on the structural difference between hyperforin and the metabolites in this study is welcome.

Response: We thank the reviewer for this suggestion. We added Supplementary Fig. 25 to give an overview of the PPAP scaffolds typically found in *H. perforatum*. In the final chapter of the manuscript, we focus on comparing the specific activities of *H. sampsonii* and *H. perforatum* CPTs which give rise to the different PPAP structures found in the plants. The following statements are included in the revised manuscript.

“PPAPs with aliphatic acyl groups, such as hyperforin and its derivatives, are renowned constituents of *Hypericum perforatum* (St. John's wort, Supplementary Fig. 25), but uncommon in *H. sampsonii*.”

“In agreement with the isolated metabolites, the *H. sampsonii* enzyme catalysed the formation of 7-*epi*-secohyperforin (*endo* configuration), whereas the *H. perforatum* enzyme yielded secohyperforin (*exo* configuration).”

“... HsCPTa directed the initial prenyl transfer to the C-3 prenyl group of the precursor, whereas HpCPTa1 prenylated the C-3 geranyl group to form hyperforin.”

Reviewer #1 (Remarks to the Author):

The authors have well addressed my concerns, and the manuscript can be accepted in the present version. Congratulations!

Reviewer #2 (Remarks to the Author):

The updated manuscript has addressed all my previous points. The additional identification of HsCPTa's role in producing an endo diastereomer brought new insights to our understanding of these CPTs. The identification and characterization of the HpCPTa1 also closes a gap for hyperforin biosynthesis. I extend my congratulations to the authors for delivering a remarkable contribution to the field.